# Shift of Host Range for the Immature Stages of the Lanternfly, *Pyrops watanabei* (Matsumura) (Hemiptera, Fulgoridae) Native to Taiwan

**DOI:** 10.3390/insects13090826

**Published:** 2022-09-12

**Authors:** Meng-Hao Hsu, Yueh-Lin Yang, Meng-Ling Wu, Liang-Jong Wang

**Affiliations:** 1Taiwan Forestry Research Institute, Council of Agriculture, Executive Yuan, Taipei City 10079, Taiwan; 2Institute of Ecology and Evolutionary Biology, National Taiwan University, Taipei City 10617, Taiwan

**Keywords:** host plants, native species, Heptapleurum heptaphyllum, Triadica sebifera

## Abstract

**Simple Summary:**

*Pyrops watanabei* is a lanternfly species native to Taiwan, and the adults are frequently on *Triadica sebifera* in summer. Compared to adult longevity, the developmental duration of immature stages from eggs to adult emergence is much longer. Although few records exist, the plants preferred by the immature stages had not been determined prior to this study. Thus, a one-year investigation was conducted to verify the oviposition site preference, determine the plants preferred by nymphs, and examine the change of host ranges with time and development. We establish that *Heptapleurum heptaphyllum* is not only the preferred egg-laying site, but the main host plant for nymphs from September to the next April, according to our investigations in northern Taiwan. Moreover, the preferred host plant for the nymphs shifts to *Triadica sebifera* from May onwards to adult emergence.

**Abstract:**

Although *Pyrops watanabei* is a species native to Taiwan, many fundamental aspects of the species are still poorly documented. Populations of the lanternfly in locations of northern Taiwan were found in abundance and were suitable for the conduction of an investigation from 1 July 2021 to 30 June 2022. We established the shift in the main host plants with different developmental stages. The occurrence of immature individuals on *Heptapleurum heptaphyllum* increased with age from eggs to nymphs in the third instar; however, it declined from the fourth instar onwards due to a shift in preference to *Triadica sebifera.* In 2021, the earliest detection of an egg mass was on 1 July. More eggs were recorded in August, and some could be found in September and October of the same year. In 2022, we found an egg mass on 28 June. In August 2021, nymphs in the first and second instars were detected. Then, nymphs in the third and fourth instars appeared after September and October 2021, respectively. Furthermore, nymphs in the fifth instar were sighted as late as March 2022. Finally, this study will provide a basis for further evaluation of the impact of invasion of *Pyrops candelaria* on the ecology of *Pyrops watanabei*.

## 1. Introduction

In Taiwan, *Pyrops watanabei* was listed as a protected species in 1989 and was removed from the list in 2009 [1,2]. However, many fundamental aspects of the knowledge of the species remain uncertain, such as its host plants, nymphal development, etc. [3]. The adults are frequently recorded on *Triadica sebifera* and sometimes on *Triadica cochinchinensis* in summer [2,3,4]. Only scarce records of the immature stages have been reported [2,3,4], so their overwintering ecology is unclear. The studies on *P. watanabei* have mainly been on the taxonomy [3,5] or ecology [2,3,4] of its adults, with little mention of its immature stages. Two documents reported five nymphal instars of *P. watanabei* without a description of their host plants [2,4]. Constant and Pham (2017) [3] reported two records of nymphs, one on *T. sebifera* and the other on an undetermined host. Only some records on the plants used for laying eggs have been reported [2,4]. 

Our recent results revealed that the longan is the key host for adults of another lanternfly, *P. candelaria,* whereas *T. sebifera* is most preferred by its nymphs [6]. Therefore, we think that plant species other than *T. sebifera* and *T. cochinchinensis* probably serve as the main developing hosts for the immature stages of *P. watanabei*. Furthermore, knowledge of host plants for immature stages may be conducive to explaining why some locations, e.g., green areas in cities or wetlands, are rich in *T. sebifera*, but with only scarce *P. watanabei*.

The populations of this native species, mainly distributed in northern Taiwan [2,4], have possibly been under threat from the spread of the invasive longan lanternfly, *P. candelaria*, since 2018 [6,7]. Moreover, the coexistence of both native and invasive lanternfly species on *T. sebifera* may complicate the decision of a control strategy for the latter species. In this article, we attempt to verify the differences in occurrences and habitats between these two lanternfly species for the application of possible control measures on the same target tree species. 

## 2. Materials and Methods

### 2.1. Study Sites

Investigations were conducted in the cities Taoyuan, New Taipei, and Taipei in northern Taiwan from 1 July 2021 to 30 June 2022. Study sites were chosen because abundant adults of *Pyrops watanabei* were observed on *Triadica sebifera* or *Triadica cochinchinensis*, and we could find eggs and nymphs in the nearby areas. Detailed profiles of these study sites are listed in Table 1.

### 2.2. Insect and Plant Recording

Investigations were conducted at least twice every week, and the first egg mass was detected on 1 July 2021. The plant species and inanimate objects were recorded on which immature individuals were detected. For the scientific names of the species and family of plants used in this study, we referred to the Catalogue of Life in Taiwan, a website http://taibnet.sinica.edu.tw/ (accessed on 11 August 2022) maintained by the Center for Digital Cultures and Biodiversity Research Center, Academia Sinica, Taiwan [8]. Moreover, to avoid confusion of the egg masses of the two lanternfly species, only unhatched ones were recorded. Compared with the egg mass of *Pyrops candelaria*, usually covered by a film of white wax, the unhatched egg mass of *Pyrops watanabei* is always completely covered by thick layers of white wax (Figure 1). Furthermore, for hatched egg masses with little or no wax remaining, it cannot be determined whether they were laid this year. The count of immature individuals on each plant or object was recorded. We photographed or collected some of the egg masses and nymphs. Moreover, we could distinguish the developing instars by classifying them into five distinct groups, differing in size and color, and could thus count the number of each instar (Figure 2). In June, most of the individuals had developed into adults, but some of them were pink and still unable to fly (Figure 3). Therefore, this one-year study was ended on 30 June 2022. Besides this, to avoid confusion between the nymphs of *Pyrops candelaria* and *Pyrops watanabei* during the investigation, we carefully identified the species based on some conspicuously morphological traits [2]. On the terga, nymphs of *Pyrops watanabei* have distinct light patterns that are useful for telling them apart from those of *Pyrops candelaria*. Observation was usually conducted in the woods, so it was carried out with a highly luminous flashlight (F1R or MT 14, Ledlenser GmbH & Co. KG, Solingen, Germany). A telescope (Pentax Papillio II, Ricoh imaging Co. Ltd., Tokyo, Japan) was used while the targets were high on the trees.

## 3. Results

### 3.1. Oviposition Site Preference

A total of 186 egg masses were detected on 174 living and non-living objects, including 161 plants, one cement pot, 11 dead trees, and one wooden wall of a farmhouse during the period from 1 July to 6 October 2021 and on 28 June 2022. The host plants belonged to 38 species of 26 families, and all of them were woody plants. Approximately 21.0%, 14.0%, and 8.1% of the egg masses were found on *Heptapleurum heptaphyllum, Acacia confusa,* and *Machilus thunbergii,* respectively. Only 6.5% and 2.2% of the egg masses were recorded on *Triadica sebifera* and *Triadica cochinchinensis*. Two egg masses were detected on longan trees, *Dimocarpus longan*. An egg mass was found on *Juniperus chinensis*, belonging to Gymnosperm. Only 7.5% of the egg masses were recorded on shrubs, belonging to seven species: *Psychotria rubra, Murraya exotica, Ilex asprella, Clerodendrum cyrtophyllum, Magnolia coco, Duranta erecta,* and *Lantana camara*. No egg masses were recorded on any herbaceous plants (Table 2). 

### 3.2. Host Range for Nymphs

A total of 3307 nymphs were detected on 1393 plants, consisting of 1311 trees and 82 shrubs, from 1 July 2021 to 30 June 2022. These plants were classified into 19 tree, 11 shrub, and 3 liana species of 22 families. More than 92.0% of the nymphs were recorded on trees. Nearly 65.5% of the nymphs were recorded on *Heptapleurum heptaphyllum,* and 18.4% were recorded on *Triadica sebifera.* Moreover, on these two tree species, all nymphal stages were recorded. Nymphs in the final, fifth instar were also detected on an additional two tree species, *Triadica cochinchinensis* and *Acacia confusa*. The third-highest number of nymphs detected on plants was on *Callicarpa formosana,* a shrub species. Most of the averages were no more than 10 nymphs per plant, except for the records of *Mallotus paniculatus*, *Diospyros morrisiana,* and *Dimocarpus longan* because of the observation of newly hatched first instars beside an egg mass (Table 3). 

### 3.3. Shift in Host Range Based on Time

To determine whether the host range changed over time, we calculated and divided the monthly percentage of the occurrence of nymphs into three groups of plants: *Heptapleurum heptaphyllum*, *Triadica sebifera,* and other plants. In Figure 4, a growing trend in the host preference towards *Heptapleurum heptaphyllum* was apparent with time from August to December 2021, and the percentage occurrence was still high until May 2022. In contrast, the percentage occurrence of nymphs on *Triadica sebifera* began to rise after winter from February to June 2022. In July 2021 and in May and June 2022, the occurrences of nymphs on *Triadica sebifera* were higher than those on *Heptapleurum heptaphyllum.* The occurrence of nymphs on plants other than the two above-mentioned species was highest only in August 2021.

### 3.4. Shift in Host Range Based on Developmental Stage 

As indicated in Figure 5, at least three remarkable trends in host plant preference according to different developmental stages can be noticed. First, more than 70% of the egg masses were recorded on the group of non-living objects and plants other than *Heptapleurum heptaphyllum* and *Triadica sebifera.* Second, individuals of the final, fifth instar were most likely found on *Triadica sebifera.* Lastly, nymphs in the second, third, and fourth instars apparently preferred *Heptapleurum heptaphyllum* as their host plant. In addition, the first instar seemed to be a transition period.

### 3.5. Monthly Records of Egg Masses and Nymphs in Different Instars

In 2021, we began to detect the egg mass on 1 July, and 139 egg masses were observed in August. However, some egg masses were oviposited in September, and the last one of the same generation was recorded on 6 October. The next generation began on 28 June 2022, as indicated by the detection of a newly laid egg mass. Nymphs were found on plants monthly during the entirely one-year investigation period, and nymphs in the first instar began to be detected on 14 August 2021. Only the first and second instars of nymphs were observed in August. Then, the third and fourth instars began to appear in September and October, respectively. The nymphs that survived during winter months, i.e., from December 2021 to February 2022, were mainly in the third instar, whereas many nymphs in the second and fourth instars could also be found. The fifth instar was recorded from March 2022 onwards (Table 4).

## 4. Discussion

This is the first report indicating that *Heptapleurum heptaphyllum,* which is distributed in thickets and secondary forests on hills and mountains at low elevations throughout Taiwan [9,10], is the key host plant for the immature stages of *Pyrops watanabei* based on our investigations in the cities of northern Taiwan. On *H. heptaphyllum*, the eggs and all nymphal stages were detected. Especially during winter months, the preference for *H. heptaphyllum* is probably due to plant phenology. Compared to deciduous *Triadica sebifera* [11]*, H. heptaphyllum* is semi-deciduous or evergreen [11] and could thus be a shelter for nymphs in winter. Furthermore, it probably also supplies better tree sap rich in nutrients for nymphs. Based on a survey in Taiwan, the florescence of *H. heptaphyllum* occurs around December and January, and it is one of the few sources for winter honey in Taiwan markets [12]. Compared to deciduous trees, the evergreen ones are more likely to be shelters during winter for many insect species in the forms of adults or immature stages. For example, in Taiwan longan trees are winter shelters used by two adult hemipterans, *P. candelaria* (unpublished data, Hsu) and the litchi stink bugs, *Tessaratoma papillosa* [13]. It is similar to the cases in *P. watanabei* and *P. candelaria*, the spotted lanternfly, *Lycorma delicatula,* has growing preferences for one of its host plants, *Ailanthus altissima,* with developmental stages [14]. However, unlike the two lanternfly species in Taiwan, *L. delicatula* survives the winter as eggs from September to the next May in countries with a colder climate [15].

In this study, we found that the host plant preference of the final nymphal stage is different from those of younger ones. Most of the nymphs in the fifth instar were recorded on *T. sebifera* rather than *H. heptaphyllum,* especially from May onwards. Although we frequently detected adults—especially the newly emerged, pink ones yet incapable of flight—on *T. sebifera* and *T. cochinchinensis*, no adult was observed on any *H. heptaphyllum* (unpublished data, Hsu). Therefore, we assume that in March, nymphs begin to gradually leave *H. heptaphyllum* for *T. sebifera*; in spring, the occurrence of nymphs is increasingly higher monthly, while more new branches and leaves grow on the latter, deciduous host plant. Finally, adult emergence is more likely to take place on *T. sebifera* in summer from late May to early July. Therefore, according to Constant and Pham (2017) [3], the record of a fifth-instar nymph on *T. sebifera* was in June, and observations of many adults were made in August on the same plant species. 

In summer, *Pyrops watanabei* lays eggs approximately one month after feeding on *T. sebifera* [4]. During that time, the adults feed on *T. sebifera* already in bloom, and the plant is probably rich in nutrients to later grow fruits with seeds enclosed in a wax-like substance, utilized in the past for the production of traditional vegetable tallow [11]. In *Pyrops watanabei,* the species-specific criteria of plants for oviposition may not be more important than the general site criteria. Therefore, it is similar to the result of the previous study [6] on *Pyrops candelaria*, wherein egg masses were usually detected on smooth surfaces of tree trunks or non-living objects, such as dead trees, cement pots, cement pillars, and rocks (unpublished data, Hsu). Another lanternfly, *L. delicatula*, also has potential to lay eggs on a wide range of non-plant materials with smooth surfaces [15].

We found that although >65% of nymphs were recorded on *H. heptaphyllum,* all plants with egg masses or nymphs of *P. watanabei* were close to *T. sebifera* or *T. cochinchinensis*. These two *Triadica* species are the key host plants for adults. According to the investigation by Lin et al. (1958) [16], *T. sebifera* originated from China [17,18] a hundred years ago and is distributed mainly in the lowlands and hills of northern Taiwan. They also pointed out that *T. cochinchinensis* is native to Taiwan and distributed in mountains at low to medium altitudes (up to 1000 m) [16,18,19]. Whether the adult and older nymphal stages of *P. watanabei* have expanded their host plant preference from *T. cochinchinensis* to the later introduced *T. sebifera* still needs more research to address. However, the younger instars may stick to the native plant, *H. heptaphyllum,* as a main developing host plant and a shelter for overwintering. Recent investigations [2,4] indicated that the insect was mainly recorded in northern Taiwan, while some sporadic records of adults were still reported from central and southern Taiwan (unpublished data, Hsu and Wang). Further studies are necessary to verify whether the occurrence of nymphs on *H. heptaphyllum* and *T. sebifera* is as frequent there as it is in northern Taiwan. 

Based on this study, from 4 July to 13 August 2021, no nymphs of *P. watanabei* were detected, and from 14 August to the end of September, less than 2.5% of nymphs were recorded on *T. sebifera*. This may be a window of opportunity in which to remove as many nymphs of *P. candelaria* as possible before they emerge as adults with a minimal risk of disturbance to the native *P. watanabei*. Although *T. sebifera* is the key host plant, the habitats of these two lanternfly species differ. The native lanternfly has a tendency of distribution on hills with relatively few human dwellings, while the invasive one prefers lowlands, the foot of hills, and parks or graveyards on hillsides with much openness and human disturbance (unpublished data, Hsu). Therefore, to protect *P. watanabei* and for efficiency, we suggest avoiding control measures for *P. candelaria* on hills.

## 5. Conclusions

We established that *Pyrops watanabei* thrives in the hills with abundant populations of both *Heptapleurum heptaphyllum* and *Tricadica sebifera* or *Tricadica cochinchinensis.* Some areas, for example, lowlands and wetlands, that were rich in *T. sebifera*, but with no or scarce adults of *P. watanabei*, were probably because these habitats have scarce *H. heptaphyllum* and are not suitable for the nymphs. Thus, the results of this study shed light on the mystery of the insect’s life cycle. In July, the numbers of adult lanternflies began to decline on *T. sebifera* or *T. cochinchinensis,* because in August, we could find egg masses on many different plants near the above-mentioned two *Tricadica* species. After hatching, the host preferences towards *H. heptaphyllum* were increasingly stronger with time and also in later developmental stages. From September to November, most of the nymphs in the younger three instars could be found on *H. heptaphyllum*. Then, during the winter months from December to the next January, evergreen *H. heptaphyllum* around florescence may have served as a shelter for the nymphs in the second, third, and fourth instars. Later, in spring from March to May, increasingly more new branches and leaves of deciduous *T. sebifera* grew, and the older, fourth and fifth instars tended to leave *H. heptaphyllum* to congregate on *T. sebifera*. In June, several adults appeared on the same tree of *T. sebifera* with nymphs waiting for emergence. Finally, in terms of application, to avoid removing nymphs of *P. watanabei* by mistake, we suggest that possible measures to control nymphs of invasive *P. candelaria* can be taken from the middle of July to the end of September on trunks of *T. sebifera* in lowlands and parks or graveyards on hillsides. 

## Figures and Tables

**Figure 1 insects-13-00826-f001:**
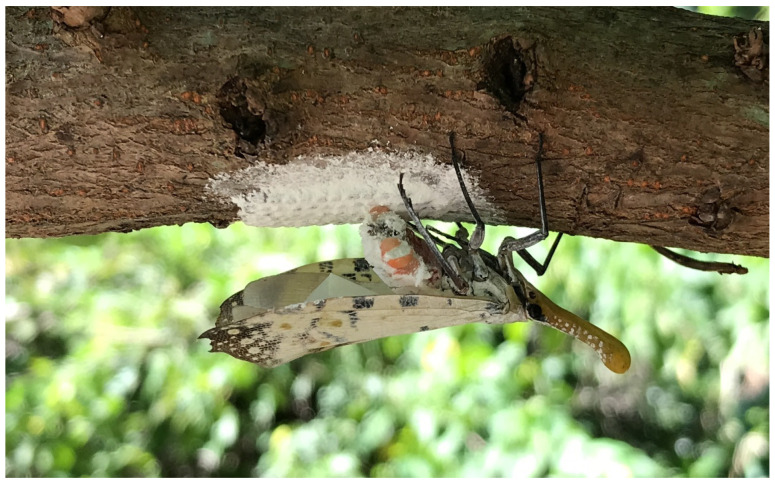
On *Triadica sebifera,* a female adult of *Pyrops watanabei* was waxing its newly laid egg mass.

**Figure 2 insects-13-00826-f002:**
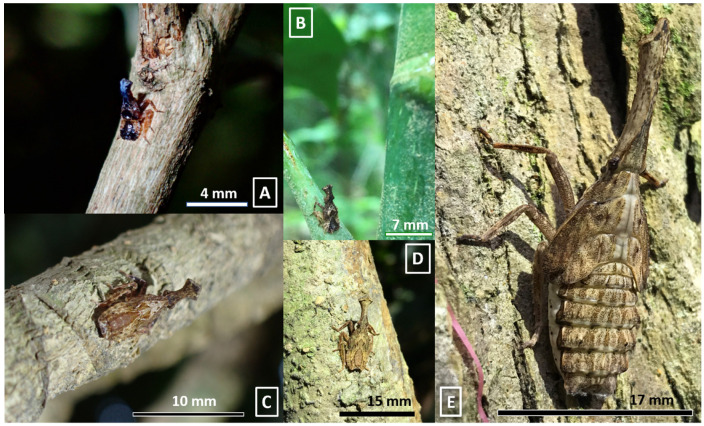
The nymphs of *Pyrops watanabei* in different instars. (**A**) A first instar on *Tibouchina semidecandra*. (**B**) A second instar on *Heptapleurum heptaphyllum*. (**C**) A third instar on *H. heptaphyllum*. (**D**) A fourth instar on *H. heptaphyllum*. (**E**) A fifth instar on *Triadica sebifera.*

**Figure 3 insects-13-00826-f003:**
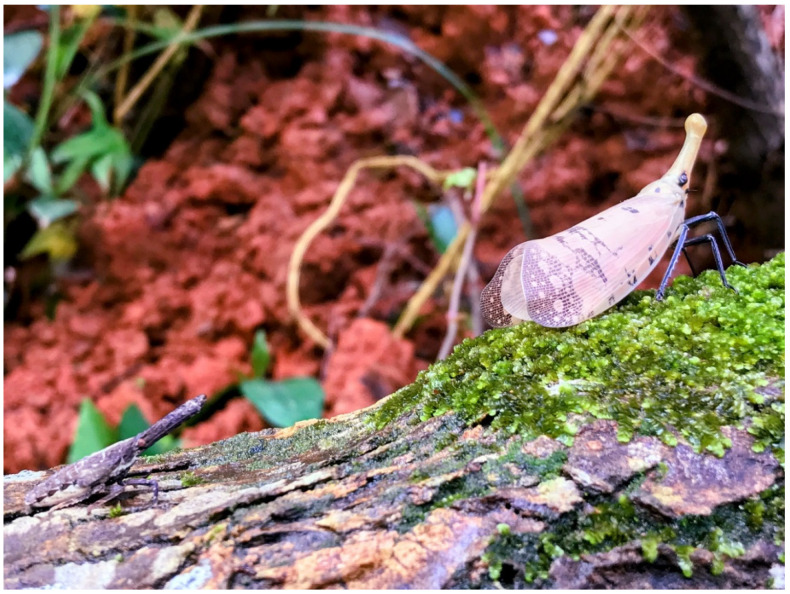
On *Triadica sebifera,* a fifth instar and a newly emerged adult of *Pyrops watanabei* with a transient pale shade of pink.

**Figure 4 insects-13-00826-f004:**
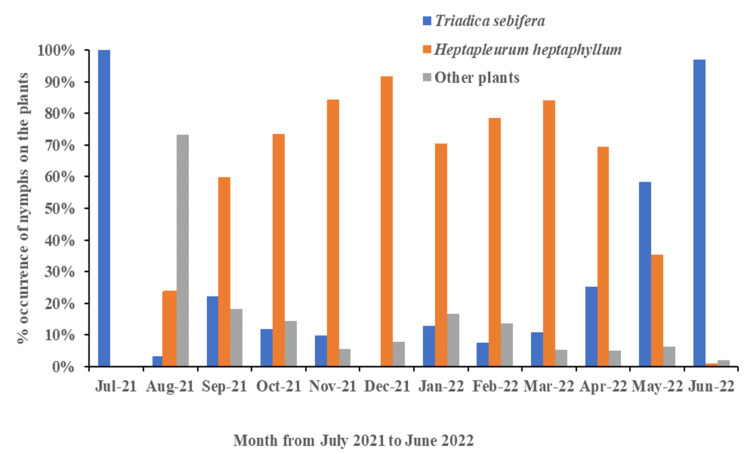
The percentage occurrence of the nymphs of *Pyrops watanabei* recorded monthly on *Triadica sebifera, Heptapleurum heptaphyllum,* and other plants from 1 July 2021 to 30 June 2022.

**Figure 5 insects-13-00826-f005:**
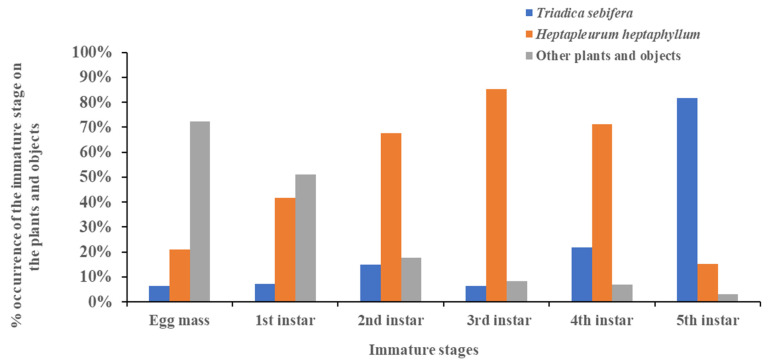
The shift of host range with different developmental immature stages from the egg mass to 5th instar of *Pyrops watanabei* recorded on *Triadica sebifera, Heptapleurum heptaphyllum*, or other plants and objects from 1 July 2021 to 30 June 2022.

**Table 1 insects-13-00826-t001:** Detailed profiles of the study sites with detected immature individuals of *Pyrops watanabei* in the cities Taoyuan, New Taipei and Taipei, Taiwan, from 1 July 2021 to 30 June 2022.

Study Sites	Profile	Elevation	Main Plants	GPS Coordinates
Mt. Danfeng, Beitou, Taipei City	A trail along the mountain ridge	120 m	*T. cochinchinensis* and *H. heptaphyllum*	25.133, 121.513
Guizikeng, Beitou, Taipei City	A greenspace on a hill	139 m	*T. sebifera* and *H. heptaphyllum*	25.152, 121.493
Mt. Tatung, Shulin, New Taipei City	A trail along the mountain ridge	237 m	*T. sebifera* and *H. heptaphyllum*	24.998, 121.409
Qinglongling, Shulin, New Taipei City	A trail on a hill	245 m	*T. sebifera* and *H. heptaphyllum*	24.999, 121.402
Shuidui Park, Wugu, New Taipei City	A park on a hillside	150 m	*T. sebifera* and *H. heptaphyllum*	25.074, 121.428
Dashuinan Rd., Linkou, New Taipei City	Green areas by the road near a golf course	255 m	*T. sebifera* and *H. heptaphyllum*	25.084, 121.341
Sinlin Trails, Linkou, New Taipei City	Trails on hills	250 m	*T. cochinchinensis* and *H. heptaphyllum*	25.066, 121.396
Mt. Dagu, Linkou, New Taipei City	Green areas by the roads near a golf course	120 m	*T. sebifera* and *H. heptaphyllum*	25.108, 121.300
Qionzaihu Temple, Taishan, New Taipei City	A trail near the temple on a hill	150 m	*T. sebifera* and *H. heptaphyllum*	25.051, 121.422
Yangchou Trails, Luzhu, Taoyuan City	Trails on hills	225 m	*T. sebifera* and *H. heptaphyllum*	25.051, 121.310
Mt. Wujiutong, Luzhu, Taoyuan City	Trails on the mountain	155 m	*T. sebifera* and *H. heptaphyllum*	25.067, 121.294
Sioucai Trails, Yangmei, Taoyuan City	Trails on hills	255 m	*T. sebifera* and *H. heptaphyllum*	24.885, 121.145
Tung Tree Trails, Guishan, Taoyuan City	Trails on hills	240 m	*T. sebifera* and *H. heptaphyllum*	25.053, 121.343

**Table 2 insects-13-00826-t002:** Plant species or objects for oviposition of *Pyrops watanabei* recorded from 1 July to 6 October 2021 and on 28 June 2022.

**Plant Species or Objects**	**Family**	***n* ^a^**	**No. Egg Masses**
*Heptapleurum heptaphyllum*	Araliaceae	38	39
*Acacia confusa*	Fabaceae	24	26
*Machilus thunbergii*	Lauraceae	15	15
Wood ^b^		12	12
*Mallotus paniculatus*	Euphorbiaceae	10	10
*Triadica sebifera*	Euphorbiaceae	9	12
*Macaranga tanarius*	Euphorbiaceae	6	6
*Machilus zuihoensis*	Lauraceae	5	5
**Psychotria rubra*	Rubiaceae	5	5
*Triadica cochinchinensis*	Euphorbiaceae	4	4
*Rhus succedanea*	Anacardiaceae	4	4
**Murraya exotica*	Rutaceae	3	3
*Liquidambar formosana*	Altingiaceae	3	3
*Semecarpus gigantifolius*	Anacardiaceae	3	3
*Ardisia sieboldii*	Primulaceae	2	3
*Lagerstroemia subcostata*	Lythraceae	2	4
*Bridelia balansae*	Phyllanthaceae	2	2
**Ilex asprella*	Aquifoliaceae	2	2
*Pouteria campechiana*	Sapotaceae	2	2
*Diospyros morrisiana*	Ebenaceae	2	2
*Dimocarpus longan*	Spindaceae	2	2
*Ficus elastica*	Moraceae	1	2
*Meliosma rigida*	Sabiaceae	1	3
*Gordonia axillaris*	Theaceae	1	1
*Elaeocarpus decipiens*	Elaeocarpaceae	1	1
**Clerodendrum cyrtophyllum*	Lamiaceae	1	1
*Ficus erecta*	Moraceae	1	1
**Magnolia coco*	Magnoliaceae	1	1
**Duranta erecta*	Verbenaceae	1	1
*Daphniphyllum glaucescens*	Daphniphyllaceae	1	1
*Morus australis*	Moraceae	1	1
*Randia cochinchinensis*	Rubiaceae	1	1
*Koelreuteria henryi*	Spindaceae	1	1
Cement pot		1	1
*Elaeocarpus japonicus*	Elaeocarpaceae	1	1
**Lantana camara*	Verbenaceae	1	1
*Syzygium jambos*	Myrtaceae	1	1
*Cinnamomum camphora*	Lauraceae	1	1
*Juniperus chinensis*	Cupressaceae	1	1
*Prunus campanulata*	Rosaceae	1	1
Sum		174	186

^a^ Only plants or objects with more than a mass of unhatched eggs were noted. ^b^ A wooden wall of the farmhouse and 11 dead trees. * Shrub species.

**Table 3 insects-13-00826-t003:** Plant species and numbers of nymphs in different instars of *Pyrops watanabei* recorded from 1 July 2021 to 30 June 2022.

Plant Species	Family	*n* ^a^	No. Nymphs						Mean
			1st Instar	2nd Instar	3rd Instar	4th Instar	5th Instar	Total ^b^	
*Heptapleurum heptaphyllum*	Araliaceae	1072	231	345	1085	454	50	2165	2.0
*Triadica sebifera*	Euphorbiaceae	188	40	76	83	139	270	608	3.2
**Callicarpa formosana*	Lamiaceae	43	9	44	50	13	0	116	2.7
*Triadica cochinchinensis*	Euphorbiaceae	16	0	2	7	16	8	33	2.1
**Maesa perlaria*	Primulaceae	15	2	11	14	2	0	29	1.9
*Acacia confusa*	Fabaceae	10	0	0	9	7	2	18	1.8
**Breynia officinalis*	Phyllanthaceae	8	20	12	1	1	0	34	4.3
**Psychotria rubra*	Rubiaceae	5	43	1	1	1	0	46	9.2
*Lagerstroemia subcostata*	Lythraceae	5	18	9	0	0	0	27	5.4
**Viburnum parvifolium*	Adoxaceae	4	0	1	1	2	0	4	1.0
*Ilex ficoidea*	Aquifoliaceae	3	0	0	7	0	0	7	2.3
*Machilus thunbergii*	Lauraceae	2	0	0	8	1	0	9	4.5
*Ficus erecta*	Moraceae	2	0	3	0	0	0	3	1.5
**Melicope pteleifolia*	Rutaceae	1	0	1	0	0	0	1	1.0
**Murraya exotica*	Rutaceae	1	5	0	0	0	0	5	5.0
^†^Mussaenda parviflora	Rubiaceae	1	0	0	1	0	0	1	1.0
*Symplocos chinensis*	Symplocaceae	1	4	0	0	0	0	4	4.0
*Elaeocarpus decipiens*	Elaeocarpaceae	1	1	2	0	0	0	3	3.0
*Psidium guajava*	Myrtaceae	1	0	1	3	0	0	4	4.0
*Bischofia javanica*	Phyllanthaceae	1	5	0	0	0	0	5	5.0
^†^ *Piper kadsura*	Piperaceae	1	0	0	1	1	0	2	2.0
**Clerodendrum ohwii*	Lamiaceae	1	0	0	1	0	0	1	1.0
*Randia cochinchinensis*	Rubiaceae	1	0	1	0	0	0	1	1.0
*Pouteria campechiana*	Sapotaceae	1	1	0	0	0	0	1	1.0
*Archidendron lucidum*	Fabaceae	1	0	1	1	1	0	3	3.0
^†^ *Zanthoxylum nitidum*	Rutaceae	1	0	1	0	0	0	1	1.0
*Ilex rotunda*	Aquifoliaceae	1	0	0	1	0	0	1	1.0
**Ilex aspella*	Aquifoliaceae	1	3	0	0	0	0	3	3.0
**Tibouchina semidecandra*	Melastomataceae	1	10	0	0	0	0	10	10.0
**Itea parviflora*	Iteaceae	1	10	0	0	0	0	10	10.0
*Mallotus paniculatus*	Euphorbiaceae	1	32	0	0	0	0	32	32.0
*Diospyros morrisiana*	Ebenaceae	1	50	0	0	0	0	50	50.0
*Dimocarpus longan*	Spindaceae	1	70	0	0	0	0	70	70.0
Sum		1393	554	511	1274	638	330	3307	

^a^ Only plants with more than one nymph were counted. ^b^ Total numbers of nymphs of all instars. * Shrub species. ^†^ Liana (woody vine) species.

**Table 4 insects-13-00826-t004:** Monthly records of egg masses and nymphs of *Pyrops watanabei* from 1 July 2021 to 30 June 2022.

Time		No. Egg Masses	No. Nymphs					
Year	Month		1st Instar	2nd Instar	3rd Instar	4th Instar	5th Instar	Total ^a^
2021	July	8	0	0	0	2	15	17
	August	139	306	5	0	0	0	311
	September	34	184	125	12	0	0	321
	October	4	42	125	107	1	0	275
	November	0	20	76	127	8	0	231
	December	0	0	61	262	65	0	388
2022	January	0	2	55	224	81	0	362
	February	0	0	43	293	93	0	429
	March	0	0	19	203	118	4	344
	April	0	0	2	46	189	94	331
	May	0	0	0	0	47	49	96
	June	1	0	0	0	34	168	202
Sum		186	554	511	1274	638	330	3307

^a^ Total numbers of nymphs of all instars.

## Data Availability

The data presented in this study are available on request from the corresponding author.

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
