# Peer review of "Shift of Host Range for the Immature Stages of the Lanternfly, Pyrops watanabei (Matsumura) (Hemiptera, Fulgoridae) Native to Taiwan"

_insects, 2022, doi:10.3390/insects13090826_

Round 1

Reviewer 1 Report

The lanternfly, Pyrops watanabei was once listed as a protected species in Taiwan, but knowledge of its life history is poorly documented. This manuscript provides new and valuable biological information of this species based on a one-year intensive survey in northern Taiwan. It is the first report indicating Heptapleurum heptaphyllum is the main host plant for the immature stages (eggs and all nymphal stages) of P. watanabei. The results also got supported evidence to reveal that host plant preference shifted with time and developmental stage.

Overall the manuscript is well-written with high scientific merit and does reach the publishing quality of Insects. I recommend the acceptance of the manuscript after a minor revision.

Some comments are listed below which the authors may consider to improve the manuscript.

1.      Do not cite the Table or Figure number in parentheses at the end of subheading in the Results section. See Lines 123, 140, 154, 165, 174.

2.      Consider grouping Figures 2-6 in a single figure (composite figure), accompanied with scale on each figure, for ease of comparison of instars.

3.      Figure 8: Change the Chinese date (month/day) in the axis of abscissa to English.

4.      References written in Chinese or Japanese should be noted in parentheses at the end of each reference.

5.      Other comments please see the marks and corrections made in the attached pdf file.

Author Response

Dear sir,

We took your excellent advice and combined five figures into one picture with scales on each instar for comprehension by readers. Other comments have been taken and revised in this new manuscript.

Thank for your comments, we hope the combined figure will be better and suitable for publication.

Reviewer 2 Report

Overall the paper is adequately designed and executed.  Some grammar can be improved.  I think the paper would benefit for reference to the literature on Fulgoridae, related to host range shifts during development, and egg-laying sites.

Line 14 – had not been determined prior to this study.

Line 21 – is a species native to Taiwan.

Line 22 – delete “of the knowledge”

Line 27-28 – due to a shift in preference to Triadica sebifera

34-35 – The results of this study will provide a basis for further evaluation of the impact of invasive Pyrops candeluria on the ecology of Pyrops watanabei.

45 – so their overwintering ecology is unclear.

49-50 two records of nymphs, one on T. sebifera, the other on an undetermined host.

56 – delete “the discussion of”

59-65 – this paragraph would flow better if moved before the previous paragraph.

66-67 – you close your Introduction with a statement of results.  Should place in abstract or results section.  Delete here.

82 – “when” should be “on which immature individuals were observed.”

88 -  spread should read covered.

Footnote for Table 3 – Only plants with more than one nymph were counted.

55-156 – you are looking to see if host range changes over time.  This happens with Lycorma delicatula, another fulgorid.  Is this common in the family?  If so, it would bear further comment here or elsewhere.

167 – laying eggs on inanimate objects as well as host plants also common in L. delicatula.

189-190 – which occurs in thickets…  Better to say low elevation. Altitude better used to describe height above ground.

193-194 – saying the preference for a host is due to plant physiology is a pretty vague statement.  Can you be more specific?

195 – some other insects use winter shelter plants – can you expand on this?

Aside from the grammatical points mentioned, I believe the paper will benefit from discussion relating your results to insects in this family in general.

Author Response

Dear sir,

In this revised version, we took your good advice and added the wordings in “Discussion” on (1) some inanimate objects are oviposition sites of Lycorma delicatula, (2) host range of L. delicatula changes over time, and (3) other insects such as two hemipterans, Pyrops candelaria and Tessaratoma papillosa, use winter shelter plants.

We changed “the preference for a host is due to plant physiology” to “….phenology”. Phenology means trees can be classified as evergreen or deciduous, and their florescence are in spring or in winter according to their species. All of other comments on wording are accepted.

We added additional three references on host preference or oviposition sites of other hemipteran insects, including P. candelaria, L. delicatula, and T. papillosa, for comparison to our target insect, P. watanabei. Thank you for our useful comments.